# Exploring the Synergistic Effect of Bergamot Essential Oil with Spironolactone Loaded Nano-Phytosomes for Treatment of Acne Vulgaris: In Vitro Optimization, In Silico Studies, and Clinical Evaluation

**DOI:** 10.3390/ph16010128

**Published:** 2023-01-15

**Authors:** Rofida Albash, Noha M. Badawi, Mohammed I. A. Hamed, Maha H. Ragaie, Sahar S. Mohammed, Rovan M. Elbesh, Khaled M. Darwish, Manar O. Lashkar, Sameh S. Elhady, Shaimaa Mosallam

**Affiliations:** 1Department of Pharmaceutics, College of Pharmaceutical Sciences and Drug Manufacturing, Misr University for Science and Technology, Giza P.O. Box 77, Egypt; 2Department of Pharmaceutics and Pharmaceutical Technology, Faculty of Pharmacy, The British University in Egypt, Cairo 11837, Egypt; 3Department of Organic and Medicinal Chemistry, Faculty of Pharmacy, Fayoum University, Fayoum 63511, Egypt; 4Department of Dermatology, STD’s and Andrology, Faculty of Medicine, Minia University, Al-Minya 61111, Egypt; 5Department of Physical Therapy for Women’s Health, Faculty of Physical Therapy, Misr University for Science and Technology, Giza P.O. Box 77, Egypt; 6Department of Medicinal Chemistry, Faculty of Pharmacy, Suez Canal University, Ismailia 41522, Egypt; 7Department of Pharmacy Practice, Faculty of Pharmacy, King Abdulaziz University, Jeddah 21589, Saudi Arabia; 8Department of Natural Products, Faculty of Pharmacy, King Abdulaziz University, Jeddah 21589, Saudi Arabia; 9Department of Pharmaceutics and Industrial Pharmacy, Faculty of Pharmacy, October 6 University, Giza 12585, Egypt

**Keywords:** bergamot essential oil, spironolactone, nano-phytosomes, acne vulgaris, in silico study, clinical studies, drug discovery, industrial development

## Abstract

The foremost target of the current work was to formulate and optimize a novel bergamot essential oil (BEO) loaded nano-phytosomes (NPs) and then combine it with spironolactone (SP) in order to clinically compare the efficiency of both formulations against acne vulgaris. The BEO-loaded NPs formulations were fabricated by the thin-film hydration and optimized by 3^2^ factorial design. NPs’ assessments were conducted by measuring entrapment efficiency percent (EE%), particle size (PS), polydispersity index (PDI), and zeta potential (ZP). In addition, the selected BEO-NPs formulation was further combined with SP and then examined for morphology employing transmission electron microscopy and three months storage stability. Both BEO-loaded NPs selected formula and its combination with SP (BEO-NPs-SP) were investigated clinically for their effect against acne vulgaris after an appropriate in silico study. The optimum BEO-NPs-SP showed PS of 300.40 ± 22.56 nm, PDI of 0.571 ± 0.16, EE% of 87.89 ± 4.14%, and an acceptable ZP value of −29.7 ± 1.54 mV. Molecular modeling simulations showed the beneficial role of BEO constituents as supportive/connecting platforms for favored anchoring of SP on the Phosphatidylcholine (PC) interface. Clinical studies revealed significant improvement in the therapeutic response of BEO-loaded NPs that were combined with SP over BEO-NPs alone. In conclusion, the results proved the ability to utilize NPs as a successful nanovesicle for topical BEO delivery as well as the superior synergistic effect when combined with SP in combating acne vulgaris.

## 1. Introduction

Acne vulgaris is a prevalent dermatological disease that affects more than 80% of adolescents. It is commonly spread in the areas of the pilosebaceous gland (oil glands accompanied by hair follicles on the face, neck, shoulders, chest, and back) [1]. In the aspect of its clinical manifestations, it is depicted by the existence of various types of skin lesions which are inflammatory or non-inflammatory and sometimes turn out to be more severe showing papules, pustules, nodules, or cysts [1]. Remarkably, it is caused by altered keratinization, androgen-induced sebum, inflammation, and Propionibacterium acnes (*P. acne*) colonization [2]. Acne vulgaris is not a life-threatening disease, even though it could have a massive influence on patients’ psychosocial and physical health. Consequently, many researchers are concerned with evading acne vulgaris [2]. The first-line treatment of acne is topical application of medications which are composed of synthetic compounds or their mixture with oral antibiotics. Nevertheless, many of these topical medications frequently lead to various shortcomings such as dryness, irritation, scaling, and itching. Accordingly, the utilization of medicinal plants has been considered as a replacement approach to overcome the drawbacks of synthetic compounds [3].

Bergamot (*Citrus medica* L. var. sarcodactylis) has been utilized as a medicinal plant because of its antifungal, stomachic, and bacteriostatic actions [4]. Bergamot peel is applied to produce bergamot essential oil (BEO) [2]. It was stated that BEO could show some effects such as immunomodulatory and anti-inflammation [5]. In addition, it was reported that BEO could treat acne vulgaris, which is triggered by excessive secretion of androgen through decreasing the sebaceous gland growth-rate, avoiding triglyceride accumulation and inflammatory cytokines (IL-1α) release in the sebaceous gland as well as promoting apoptosis in the sebaceous gland. It was stated that BEO may enhance acne lesions through suppressing *P. acnes* and improving inflammatory response [2].

However, one of the main challenges of phytochemicals (naturally occurring constituents from plants) is their poor penetration via the skin barrier which might prohibit their dermal administration [3]. Nanotechnology application has been investigated by many researchers to resolve the problems of topical application of natural products [6]. Nanocarriers can enhance the penetration of the entrapped drugs by improving their solubility and protecting them against instability problems [7]. Lipid nanocarriers are frequently used to accomplish such purposes [3]. Among the latest lipid nanocarriers are nano-phytosomes (NPs), which are made by mixing phytoconstituents with phosphatidylcholine (PC) that will produce a complex with stronger bonds [8]. The existence of chemical bonds between the encapsulated drug and PC offers advantages such as a better stability profile, high entrapment efficiency (EE%), and increased absorption leading to enhanced bioavailability and pharmacological activity [9].

Spironolactone (SP), as an anti-androgen agent, has been evidenced successful in decreasing the sebum discharge rate in several clinical assessments [10]. However, SP when taken orally is badly absorbed from the gastrointestinal tract and showed endocrine shortcomings that limited its application due to its poor bioavailability. Therefore, topical application of SP can permit high drug levels at the site of action which in line can reduce the systemic side effects and also enhance patient compliance [10]. Previous work stated the potency of using SP-loaded nanocarriers for the effective management of acne vulgaris [11].

Therefore, in the current study, the primary aim was to load BEO as a promising phytochemical on NPs vesicles, characterized and optimized. Optimization was performed via 3^2^ full factorial design in which PC amount (X_1_) and cholesterol amount (X_2_) were chosen as independent factors, while entrapment efficiency percent (EE%; Y_1_), particle size (PS; Y_2_), and zeta potential (ZP; Y_3_) were selected as dependent responses. In addition, a secondary aim was induced which was combining the optimized BEO-loaded NPs with SP for topical delivery. Moreover, both formulations were clinically investigated against acne patients after the molecular modelling simulation study. We proposed that SP could evoke a synergistic strategy with BEO to achieve a promising candidate for acne treatment.

## 2. Results and Discussion

### 2.1. Analysis of Full Factorial Design

Nine runs were developed following the experimental design. The model selected was 2FI. The composition of the formulations and the observed responses of the design are presented in Table 1. It was observed that the predicted R^2^ values were in good harmony with the adjusted R^2^ values in all examined responses as shown in Table 2. The adequate precision value is for verifying the model’s adequacy to navigate the design space in which a ratio above 4 is suitable, and that was detected in all studied responses (Table 2).

### 2.2. Effect of Formulation Variables on the Observed Responses

#### 2.2.1. EE%

EE% is one of the essential constraints to estimate the system’s attainment to entrap the drug [12]. The EE% of BEO-NPs ranged from 37.68 ± 0.97% to 81.89 ± 1.25% (Table 1). In the present investigation, it was detected that the PC amount (X_1_) significantly (*p* < 0.0001) influenced the EE% of BEO (Figure 1A). This could be attributed to PC surface-active properties that would improve the development of strong coherent layers around the vesicles and thus can reduce the leakage of BEO [13]. Additionally, increasing the viscosity of the system could be probable at a higher PC amount that may retard the BEO external diffusion [13]. Concerning cholesterol amount (X_2_), it was recognized that increasing the amount of cholesterol caused a significant increase in the EE% (*p* < 0.0001) (Figure 1A). Cholesterol was incorporated into the vesicles to form more rigid ones via packing the gaps among the constructed PC within the bilayers of the NPs to form a more ordered membrane and then the BEO leakage will be reduced [14].

#### 2.2.2. PS and PDI

Fabrication of vesicles with optimum PS is important for drug delivery through the skin [12]. The average diameter of nanoparticles is one of the most crucial parameters for the hair follicle pathway which serves as a favorable pathway to improve the drug skin penetration for both dermal and transdermal delivery. PS could affect the depth of follicular penetration irrespective to the type of nanocarriers. It was stated that infiltration of follicular drugs developed by nanotechnology-based formulations could be enhanced by producing small-size particles to enlarge the surfaces with the stratum corneum [15]. Follicular targeting could be a remarkable selection to treat acne vulgaris, hirsutism, and androgenetic alopecia [16].

PS of the prepared BEO-NPs lay within the nano-range from 178.58 ± 0.40 to 459.40 ± 4.09 nm, as demonstrated in Table 1 and Figure 1B. ANOVA results showed that both X_1_ and X_2_ had a significant effect on the PS of the fabricated NPs. From Figure 1B, it can be detected that increasing X_1_ significantly increased the PS (*p* < 0.0001), and this is due to PC being the major component of the vesicle wall and thus likely to increase the wall thickness and accordingly the PS [12]. It can also be explained that a high PC amount led to high BEO EE%, hence the BEO expands the space among the vesicular bilayers due to BEO inclusion in the hydrophobic zones within the vesicles so BEO-NPs with the highest EE% will attain the greatest PS [17]. Additionally, the growth of vesicle size with an increase in PC amount was in accordance with earlier findings [18,19,20]. It was also found that increasing the cholesterol amount (X_2_) significantly increases the PS (*p* < 0.0001) (Figure 1B) which may be explained by the fact that upon increasing the cholesterol amount, the bilayer hydrophobicity also increased; thus, disturbance in the vesicles membrane, accordingly led to an increase in PS as an attempt to attain thermodynamic stability. Furthermore, increasing the cholesterol amount led to an increase in the rigidity of the vesicle membrane which could result in resistance to size reduction during sonication step [21].

Regarding PDI, a measure of the size distribution homogeneity [22] in which a value of zero signifies a homogenous dispersion, whereas a value of one means a heterogeneous one. The obtained ANOVA results displayed that both X_1_ and X_2_ independent variables had no significant impact on the PDI with *p* values of 0.371 and 0.063, respectively, therefore (Figure 1C). The PDI of the measured BEO-NPs ranged from 0.408 ± 0.011 to 0.773 ± 0.007 (Table 2). It was reported that high PDI is frequently observed in vesicles fabricated by the thin-film hydration technique [17].

#### 2.2.3. ZP

The occurrence of a charge could be advantageous to maintain the stability of the formulation, as electrostatic repulsive forces tend to decrease the aggregation [14]. Upon investigation of ZP, it was found that all BEO-NPs formulations exhibited negative charges that ranged from −26.52 ± 0.51 to −28.70 ± 0.152 mV, as illustrated in Table 1 and Figure 1D. ANOVA results presented that both X_1_ and X_2_ independent variables had positive significant impacts on ZP with *p* values of 0.0001 and 0.0029, respectively. Concerning X_1_, it was previously mentioned that enrollment of PC at a high level could increase the values of ZP. It was expected that in a low ionic strength medium, the polar head group is oriented in a form that the phosphatidyl group (negative) is directed to the external part while the choline group (positive) is directed to the internal part, causing a negative charge on the surface [13]. Moreover, it was observed that X_2_ had a positive effect on ZP which may be owed to that the cholesterol molecule contains a hydroxyl group in its structure. Besides, it is a rigid molecule that aids in improving the stability and rigidity of the lipid bilayer [23]. Similar findings were reported by Salem et al. (2021) who found that the negative charges in vesicles were increased when cholesterol was increased [23]. In addition, cholesterol led to high EE% and, as earlier stated, increasing drug EE% led to increasing charges obtained by the vesicles [20].

### 2.3. Determination of the Selected BEO-Loaded NPs Formulation

The selected formulation was obtained after numerical analysis of the investigated factors using a full factorial design. The purpose was to attain BEO-NPs with the highest EE% and ZP in addition to the least PS, and PDI. Consequently, the selected formula that met these conditions was B3, which included PC and cholesterol amounts of 500 and 10 mg, respectively, in its composition. It was observed that there was a good correlation between the observed values and the predicted ones. These things considered, all the obtained responses were found to have average bias percent values which were smaller than 10%, thus verifying the high model’s reliability capability. Therefore, B3 was selected for further investigation.

### 2.4. Optimization of the Selected BEO-Loaded NPs

SP, an aldosterone antagonist with anti-androgen action, has shown favorable outcomes in the treatment of acne [11]. Therefore, the selected BEO-NPs formulation was further optimized by including SP as an anti-acne agent (BEO-NPs-SP). The optimized formula showed EE% of 87.89 ± 4.14%, PS of 300.4 ± 2.25 nm, PDI of 0.571 ± 0.16, and ZP of −29.7 ± 0.81 mV.

### 2.5. Transmission Electron Microscopy (TEM)

Morphological analysis of the optimized BEO-NPs-SP formula is shown in Figure 2. The TEM image displayed that the vesicles of the optimized formula had a uniform distribution with a spherical shape.

### 2.6. Stability Study

The visual assessment of the stored optimized BEO-NPs-SP showed that neither sedimentation nor aggregation was observed throughout the storage time. Moreover, PS, PDI, ZP, and EE% values of the stored optimized formula were 283.4 ± 2.96 nm, 0.548 ± 0.01, −31.5 ± 1.64 mV, and 86.46 ± 1.72%, respectively, which verified that the alteration from the freshly prepared system was insignificant (*p*  >  0.05).

### 2.7. Molecular Modelling Simulations

Molecular docking of SP showed extended conformation and preferential orientation of the drug’s tetracyclic skeleton towards the polar phosphate head of the PC scaffold (Figure 3). Despite the lack of relevant polar interaction between SP and PC molecule, the polar oxygen functionalities were directed towards the PC phosphate moiety for partial satisfaction of the latter hydrophilic potentiality. On the other hand, the less polar thioacetyl group of the drug showed favored extension along the PC’s hydrophobic acyl chains. The depicted SP-PC binding pattern came in good agreement with reported data for several therapeutic agents, metformin and rosuvastatin, illustrating preferential anchoring towards the PC’s polar head [24,25]. Nevertheless, the absence of promising polar binding interactions could not fully compensate for the electrostatic character of PC, the thing that would impose an electrostatic penalty against SP sole binding. All of these were translated into fair binding energy of −2.59 Kcal mol^−1^.

In presence of docked NP formulation additives and BEO constituents, the docking score for the investigated SP was increased up to −7.89 Kcal mol^−1^ which highlighted their preferential role in SP-PC complex stability. Notably, the electrostatic potentiality of PC phosphate moiety was satisfied via depicted extended hydrogen bonding with COL and LIN respective free hydroxyl group (Figure 3B). Double hydrogen bonding was depicted between LIN hydroxyl group and PC’s phosphate oxygen atom (bond length/angle; 1.8 Å/148° and 2.8 Å/120°). On the other hand, the COL’s OH group served as both hydrogen bond donor and acceptor that furnished singular hydrogen bond pairing with each of PC’s P-OH (1.3 Å/171°) and LIN’s OH group (1.8 Å/150°). It is worth noting that COL’s steroidal nucleus was extended along the PC’s acyl chains in a fashion being reported for the PC-COL system [26]. The presence of cholesterol with PC was reported to extend longitudinally along the PC acyl chains preventing the free movement of the latter hydrocarbon chains and causing the system to attain a liquid-ordered lamellar phase (Lo) [26,27,28].

Concerning the docked BEO’s other constituents, they were preferentially anchored to satisfy the non-polar characters of both the PC’s hydrocarbon chains and the drug’s non-polar core skeleton. Volatile constituting fractions of BEO including LIM, BPI, and GTP depicted preferential orientation with their cage-like scaffolds at the PC’s hydrocarbon acyl chains. Additionally, BPI depicted close proximity towards the drug’s interface that would have served in beneficial hydrophobic packing of the drug’s core ring towards the PC interface. The BEO non-volatile coumarin-based components, BGR and GMC, depicted sandwich-like conformation around the docked SP structure, the thing that mediated relevant π-CH as well as van der Walls hydrophobic interactions (~4.9 Å) via the coumarin aromatic scaffold and straight hydrocarbon chains, respectively. Moreover, the coumarin’s oxygen functionality was directed towards the PC’s phosphate head for additional electrostatic satisfaction and reduced binding penalty. All together they were contributed within the enhanced SP-to-NP formulation of binding affinity.

Both the thermodynamic stability and dispersion behavior of docked BEO-NPs-SP complex were investigated within the formulation final solvent (100% water) through explicit molecular dynamics simulations. Interestingly, the BEO-NPs-SP complex showed relevant stability throughout the whole simulation run (Figure 4), while depicting average free binding energy of −93.55 ± 5.08 Kcal mol^−1^ for the SP drug towards the PC BEO-NPs system. Dominant energy contribution was assigned for the hydrophobic van der Waals potentials, while additional polar contacts were maintained throughout the entire simulation run. Following the initial 0.2 ns and until the end of the molecular dynamic simulation, the hydrogen bond pairing between COL and LIN (1.8 ± 0.17 Å/164 ± 6.62°) as well as COL and PC phosphate group (1.5 ± 0.04 Å/169 ± 4.84°) were depicted as significant for their relative stability. On the other hand, one of the initial LIN/PC polar contacts was lost across the entire simulation run owing to the inversion of the free hydrogen atom of the LIN’s hydroxyl group. Nevertheless, the latter lost polar contact was substituted with a strong hydrogen bonding furnished by COL’s hydroxyl group acting as a hydrogen bond donor (3.0 ± 0.25 Å/124 ± 4.15°). Other than polar interactions, most of the simulated BEO components depicted limited conformational/orientation altercations along the simulation timeframes. Nonetheless, only the BEO non-volatile furanocoumarins-based constituent (BRG) depicted a lateral shift for its elongated aliphatic tail in order to accommodate a closer orientation towards the SP’s steroidal nucleus. The latter conformational shift was associated with a better fixation for SP drug at the PC interface.

Monitoring the spatial configuration of PC lipophilic chains illustrated open compass-like conformation. Both COL and BEO components maintained polar contact with the PC phosphate head while being compacted at near proximity near the thing which could reason the acyl tail conformation being pulled away from each other. The later dynamic behavior would have increased the volume of the hydrophobic chain. Such type of packing allowed the BEO-NPs-SP complex to obtain an inverted cone form with a micellar-like arrangement previously reported with several small molecules [29,30] (Figure 5).

### 2.8. Clinical Evaluation of BEO-NPs and BEO-NPs-SP on Acne Patients

The recommended treatments were applied on the acne on both sides of the face for 2 months. The percent reduction for the comedones, inflammatory lesions, and total acne lesions was assessed. As demonstrated in Table 3 and Figure 6, BEO-NPs-SP formulation was topically applied on the left side’s acne lesions, revealing a percentage reduction of 100.00%, 93.31%, and 92.59% for the comedones, inflammatory lesions, and acne lesions, respectively. There was a significant reduction (*p* < 0.05) in the comedonal count, inflammatory lesions count, and the total count of acne lesions on the left side compared to the right side receiving the optimized BEO-NPs formulation, which displayed a percentage reduction of 59.17, 64.29, and 61.39 for the comedones, inflammatory lesions, and acne lesions, respectively. BEO-NPs-SP formulation was more powerful than BEO-NPs formulation on the comedones, inflammatory lesions, and total acne lesions, as noticeable from the significantly higher percent reduction for the former related to the latter (*p* < 0.05). Furthermore, neither irritation nor erythema was reported by any of the patients with any of the used formulae during the treatment (2 months). Moreover, patients reported that the formulae had retentive characteristics on the skin and were reasonably satisfactory.

Concerning the abovementioned results, both formulations were verified to be effective nano-systems for the treatment of various acne lesions. Their anti-inflammatory action which was revealed by the significant decrease in the total lesion count may be credited to the use of BEO, which shows anti-inflammatory aspects by decreasing the release of inflammatory cytokines as stated by Sun et al., 2020 [2]. Moreover, the lipid-based nanovesicles (NPs) and BEO, which are lipophilic in nature, have enhanced stratum corneum adhesion and skin deposition, therefore, permitting a better effect [16,31]. Another auspicious application of lipid-based nanovesicles is follicular drug delivery (an alternative transport pathway) for the treatment of androgenic skin disorders and hair follicles related diseases such as acne as it was informed that lipid materials may see a higher intake into hair follicles that are occupied with sebum and offer a lipophilic environment [10,16]. Additionally, the amphipathic nature of the NPs vesicles as well as their nanometer range permitted for their improved skin interaction, and advancement of their medicinal effect [7]. Lastly, BEO-NPs-SP formulation showed a better antiacne effect than BEO-NPs formulation, which could be ascribed by the occurrence of both BEO and SP which have anti-androgenic action because acne vulgaris was reported to be caused by excessive secretion of androgen [2,10]; hence, this may recommend their synergistic effect on acne lesions.

## 3. Materials and Methods

Spironolactone (SP) was supplied by SEDICO, Pharmaceuticals Co., (Cairo, Egypt). Chloroform and methanol were purchased from El-Nasr Pharmaceutical Company, Cairo, Egypt. Phosphatidylcholine (PC) from soya bean, bergamot essential oil (BEO), and cholesterol (COL) were attained from Sigma Aldrich Chemical Co. (St. Louis, MO, USA).

### 3.1. Experimental Design

The amounts of PC (100, 300, and 500 mg), and cholesterol (10, 30, and 50 mg) were chosen to be the formulation parameters (independent variables, X_1_ and X_2_), respectively. Their effect on the critical quality attributes of the developed formulation; EE% (Y_1_), PS (Y_2_), and ZP (Y_3_) were investigated using a 2-factors, 3-levels design (Table 4). Nine preparations were fabricated based on a full factorial design (Design expert^®^ software version 12).

### 3.2. Preparation of BEO-NPs

The BEO-NPs were prepared by varying both PC and cholesterol amounts via the thin-film hydration method [32]. PC, cholesterol, and BEO (300 mg) were dispersed in 10 mL of chloroform. Chloroform evaporation was conducted under a vacuum at 60 °C via a rotary evaporator (Rotavapor, Heidolph VV 2000, Burladingen, Germany). The dried film was then hydrated by adding 10 mL of distilled water for 45 min using glass beads. Finally, the prepared dispersion was sonicated for 5 min by probe sonication (Vibra Cell-Sonics Material, 130 W, 20 kHz, Newtown, CT, USA) at 40% amplitude to obtain the BEO-loaded NPs.

### 3.3. Characterization and Optimization of BEO-NPs 

#### 3.3.1. Determination of EE%

EE% of BEO-NPs was calculated using the centrifugation method. Firstly, the vesicular dispersions of the prepared formulae were subjected to centrifugation at 20,000 rpm for 1 h at a temperature of 4 °C by a cooling centrifuge (Sigma 3K 30, Germany). After that, the supernatant was separated and collected to be examined at λ_max_ 292 nm via a UV-Vis spectrophotometer (Shimadzu UV1650, Kyoto, Japan) [33]. The following equation was employed to calculate the EE% [32,34]:

EE% = ((Total BEO concentration − Free BEO concentration)/(Total BEO concentration)) × 100.

#### 3.3.2. Determination of PS, PDI, and ZP 

Measurements of PS, PDI, and ZP of the prepared formulations were achieved by using Malvern Zetasizer (Malvern Instruments Ltd., UK) via dynamic light scattering technique at 25 °C. The measurements were attained after appropriate dilution [35,36]. All assessments (EE%, PS, PDI, and ZP) were performed in triplicate ± SD.

### 3.4. Selecting the BEO-NPs Formula

The selection of BEO-NPs was based on the desirability tool, which permitted the investigation of each response at once. Choosing the BEO-NPs selected formula main principle was fabricating vesicles with the least PS, and the highest EE% and ZP. To test the model validity, the BEO-NPs selected formula was firstly prepared and then correlated to the predicted responses by calculating the bias percent [20].

### 3.5. Optimization of the Selected BEO-Loaded NPs

To optimize the selected BEO-NPs formula, spironolactone (SP) (10 mg) [11] was included during the construction of the optimized formula with BEO, PC, and cholesterol in the organic phase to produce BEO-NPs-SP. The optimized BEO-NPs-SP was further characterized for its SP EE% at λ_max_ (256) [37], while the PS, PDI, and ZP were evaluated as previously stated.

### 3.6. Transmission Electron Microscopy (TEM)

The optimized BEO-NPs-SP morphology was investigated using TEM (Joel JEM 1230, Tokyo, Japan). The NPs dispersion was deposited on a carbon-coated copper grid as a thin film, stained then observed and photographed [20,38].

### 3.7. Stability Study

The optimum formula stability was examined to investigate the degree of drug leakage, vesicles’ growth, sedimentation, or any other physical variation. The optimized formula was left in the refrigerator for three months and then evaluated by comparing the responses of the stored formula, including EE%, PS, PDI, and ZP, with the fresh formula [39].

### 3.8. Molecular Modelling Simulations

Molecular docking-coupled dynamic simulations were performed for SP on PC in presence of formulation component cholesterol as well as BEO main constituents including volatile fractions, limonene (LIM), linalool (LIN), γ-terpinene (GTP), β-pinene (BPI), and non-volatile ones, 5-geranyloxy-7-methoxycoumarin (GMC) and bergamottin (BRG) [40]. Using the AutoDock Vina 1.2.0 software suit (Scripps Research, La Jolla, CA, United States) [41,42], ligand coordinates were constructed from the isomeric SMILES strings obtained from PubChem database (PubChem_ID: 5833, 65167, 5997, 22311, 6549, 7461, 6654, 6441377, and 5471349 for SP, PC, COL, LIM, LIN, GTP, BPI, GMC, and BRG, respectively). Non-polar hydrogens and Gasteiger charges were merged, and ligand structures were transformed into pdbqt. file formats. Sequential docking flow was conducted considering PC as the target molecule and both SP and BEO components as binding ligands. The center of PC was assigned as the center of the grid box. Default parameters were set to define atom’s affinity map, as well as electrostatic and desolvation maps. Conformational search was performed via Lamarckian genetic algorithm, while a genetic algorithm was assigned for docked binding pose predictions. Ranking the obtained poses was guided through the furnished docking energy scores (binding affinity with higher negative Kcal mol^−1^). Schrödinger-PyMol (New York, NY, USA) was employed for ligand pose visualization and ligand-wise interaction analysis adopting optimum hydrogen bonding at 3.0 Å bond length and 20° bond angle cut-offs, whereas the hydrophobic/non-polar ones at ≤5.0 Å thresholds [37,43,44].

Top-docked complex was subsequently proceeded through molecular dynamics simulation under Amber-FF forcefield using the NAMD 2.13 software [45]. Complex was explicitly solvated within a 3D-cubic box (45 × 45 × 45) Å3 of TIP3P_water model and neutralized via 0.15 M NaCl final concentration under periodic boundary conditions. Constructed system of ~13,700 atoms (Table 5) was initially minimized for 100 ps before it was equilibrated at 310 K temperature under normal volume and temperature (NVT) ensemble for 500 ps. Additional equilibration stage using normal pressure and temperature (NPT) for 500 ps was conducted to prevent any artifact formations when starting the molecular dynamics production run. NVT ensemble was finally utilized in production runs for 1 ns. Average interaction potential energy between SP and PC was estimated. Snapshots at regular timeframes (0.2, 0.4, 0.6, 0.8, and 1 ns) were extracted and graphically represented.

### 3.9. Clinical Evaluation of BEO-NPs and BEO-NPs-SP on Acne patients

The clinical trial was conducted following the ethical obligations of the Declaration of Helsinki. The Institutional Board, Faculty of Medicine, Minia University has revised and accepted the research ethics committee for clinical trials (Approval No. 290-2022).

#### 3.9.1. Patients

Twenty patients (6 males and 14 females) with mild to moderate acne vulgaris, according to the grading of NilFroushzadeh et al. [46], retrieved from the Dermatology Outpatient Clinic of Minia University Hospital, were included in this split-face study. The age of the patients ranged from 15 to 32 years and the extent of their lesions ranged between 1 to 7 months. Written informed agreement was attained from each patient and the study was conducted under the guarantee of the Committee for Postgraduate and Research Studies of Minia University. Patients who did not take any treatment modalities for their acne for at least 3 months before contribution in the study were included, while pregnant and nursing patients were eliminated.

#### 3.9.2. Treatment Protocol

In this split-face study, each patient was told to use a thin film of the selected BEO-NPs formulation on the right side of the face and BEO-NPs-SP formulation on the left side of the face once at night for 12 hrs. The treatment period continued up to 8 weeks, and the patients were instructed to report any erythema or irritation encountered during treatment [47]. Patients were photographed and evaluated clinically at the baseline and after 8 weeks on both sides of the face by counting comedones, and the total number of inflammatory and acne lesions by two blinded dermatologists and the percentage reduction was calculated according to the following equation [48,49]:

Percent reduction = (No of lesions before treatment − No of lesions after treatment)/(No of lesions before treatment) × 100 (Equation (2)).

#### 3.9.3. Statistical Analysis

Mann–Whitney U test and Wilcoxon Signed Rank test were applied for nonparametric data. Student paired *t* test and student unpaired *t* test were utilized for parametric data via SPSS^®^ program 22.0. At (*p* ≤ 0.05), the difference is significant.

## 4. Conclusions

As can be observed from the aforementioned outcomes, the fabrication of BEO-loaded NPs combined with SP was recognized to be efficient in the management of acne vulgaris. BEO-NPs selected formulation was successfully prepared via the thin film hydration approach using 3^2^ full factorial design. Afterward, the selected formulation was combined with SP to emerge BEO-NPs-SP optimized formula which exhibited acceptable characteristics in terms of EE%, PS, ZP, vesicular morphology, and stability. Then, the in silico study showed the beneficial role of BEO components as helpful/connecting platforms for favored anchoring of SP on the PC interface. Furthermore, BEO-NPs-SP formula showed a significant therapeutic effect on acne patients over BEO-NPs formula. Therefore, the combination of BEO with SP was proven to have a synergistic effect in the treatment of acne vulgaris. The achieved results here suggest the newly developed combination of BEO and SP loaded in NPs is successful in mitigating the symptoms of acne vulgaris.

## Figures and Tables

**Figure 1 pharmaceuticals-16-00128-f001:**
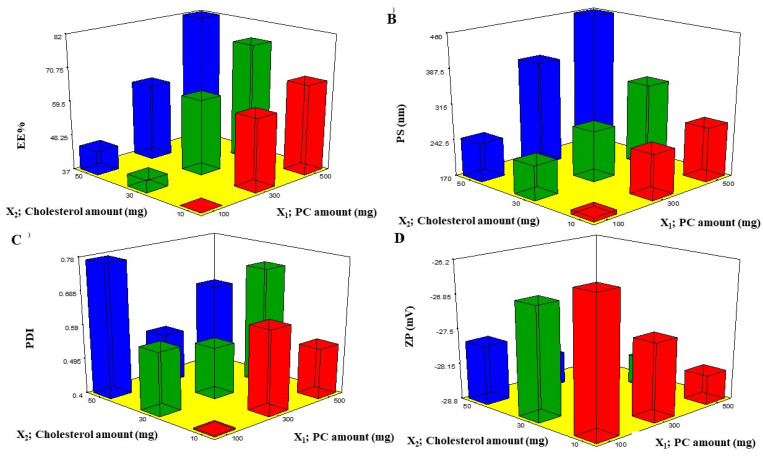
Response 3D plots (**A**–**D**) for the effect of PC amount (X_1_) and cholesterol amount (X_2_) on EE%, PS, PDI, and ZP. Abbreviations: PC: phosphatidylcholine, EE%: entrapment efficiency percent, PS: particle size, PDI: polydispersity index, and ZP: zeta potential.

**Figure 2 pharmaceuticals-16-00128-f002:**
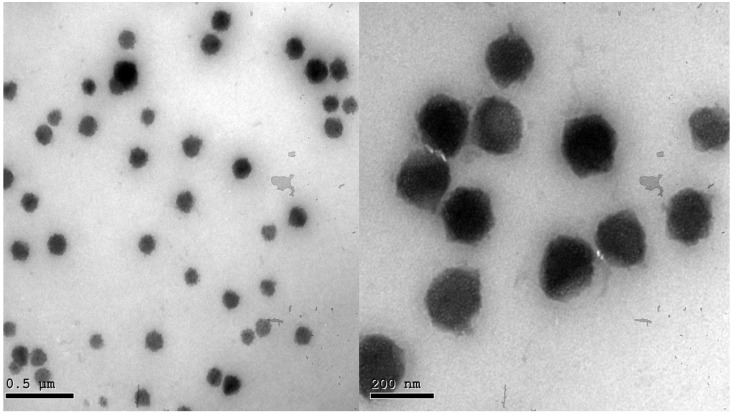
Transmission electron micrograph of the optimized BEO-NPs-SP.

**Figure 3 pharmaceuticals-16-00128-f003:**
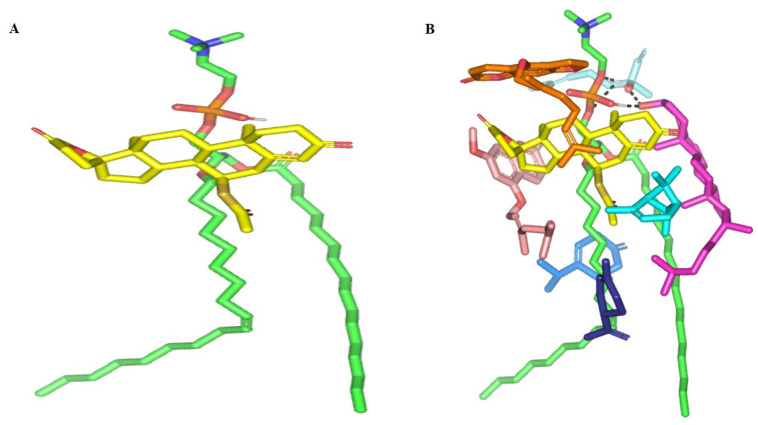
Docked binding modes of the SP-PC complexes. 3D-Stick representation SP (yellow) loaded on PC interface (green), in the absence (**A**) and in combination (**B**) with NPs formulation and BEO constituents; COL (magenta), LIN (faint blue), LIM (dark blue), GTP (marine blue), BPI (cyan), GMC (salmon), and BRG (orange). Polar (hydrogen bond) interactions are shown as black dashed lines. Abbreviations: BEO: bergamot essential oil, NPs: nano-phytosomes, SP: spironolactone, PC: phosphatidylcholine, COL: cholesterol, LIM: limonene, LIN: linalool, BPI: β-pinene, GTP: γ-terpinene, GMC: 5-geranyloxy-7-methoxycoumarin, BRG: bergamottin.

**Figure 4 pharmaceuticals-16-00128-f004:**
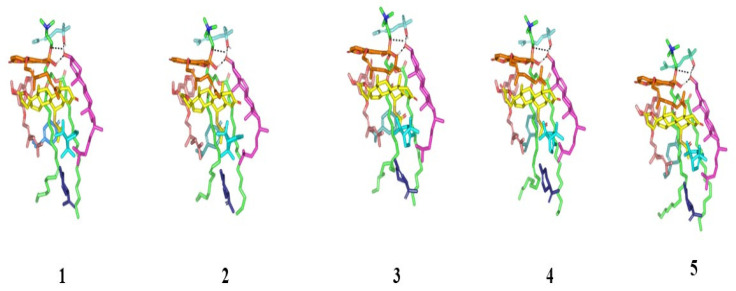
Conformational time evolution of PC-FC-BEO NPs across the all-atom molecular dynamics simulation. Thermodynamic movements of formulation components: SP (yellow) loaded on PC interface (green), COL (magenta), LIN (faint blue), LIM (dark blue), GTP (marine). Abbreviations: BEO: bergamot essential oil, NPs: nano-phytosomes, SP: spironolactone, PC: phosphatidylcholine, COL: cholesterol, LIM: limonene, LIN: linalool, GTP: γ-terpinene, 1 (0.2 ns), 2 (0.4 ns), 3 (0.6 ns), 4 (0.8 ns), and 5 (1.0 ns).

**Figure 5 pharmaceuticals-16-00128-f005:**
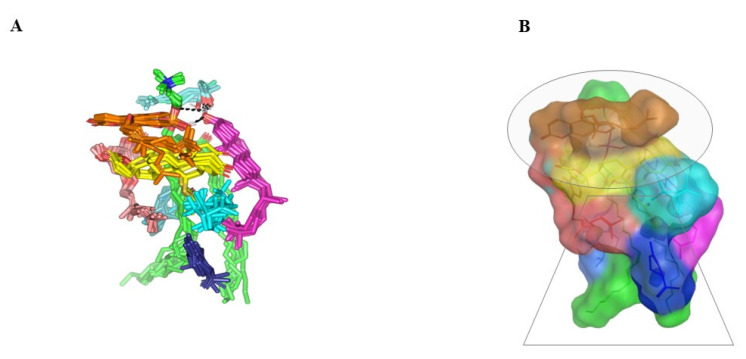
Dynamics of BEO-NPs-SP simulated formulation. (**A**) Overlay of BEO-NPs-SP complex across molecular dynamics simulation extracted frames over the initial docking conformation. (**B**) A 3D-molecular surface representation of the inverted cone micellar configuration at the end of the simulation run (1 ns). Molecules were shown in colors being previously assigned for the optimized formulation components: SP (yellow) loaded on PC interface (green), COL (magenta), LIN (faint blue), LIM (dark blue), GTP (marine blue), BPI (cyan), GMC (salmon), and BRG (orange). Abbreviations: BEO: bergamot essential oil, NPs: nano-phytosomes, SP: spironolactone, PC: phosphatidylcholine, COL: cholesterol, LIM: limonene, LIN: linalool, BPI: β-pinene, GTP: γ-terpinene, GMC: 5-geranyloxy-7-methoxycoumarin, BRG: bergamottin.

**Figure 6 pharmaceuticals-16-00128-f006:**
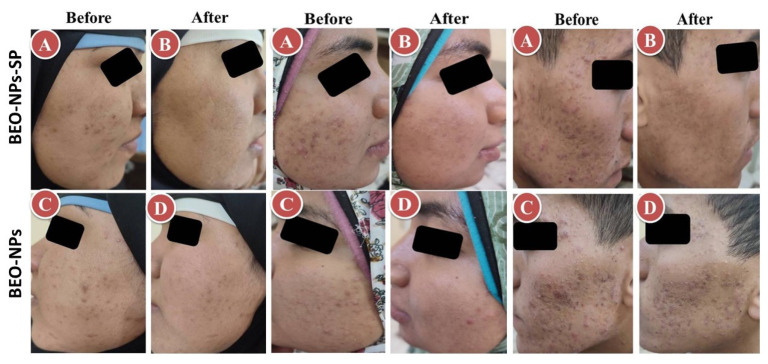
Representative patients receiving the optimized BEO-NPs-SP formulation on the left side of the face (A: Before treatment and B: After treatment) and BEO-NPs formulation on the right side of the face (C: Before treatment and D: After treatment), showing better anti-acne effect of the latter manifested by better reduction in the number of acne lesions. Abbreviations: BEO: bergamot essential oil, NPs: nano-phytosomes, and SP: spironolactone.

**Table 1 pharmaceuticals-16-00128-t001:** Experimental runs, independent variables, and measured response of the 3^2^ full factorial design for optimization of BEO loaded NPs.

Formulation Code	PC Amount (mg)	Cholesterol Amount (mg)	EE%	PS (nm)	PDI	ZP (mV)
B1	100	10	37.68 ± 0.97	178.58 ± 0.40	0.408 ± 0.011	−26.52 ± 0.51
B2	300	10	61.12 ± 1.04	262.80 ± 0.30	0.635 ± 0.063	−27.35 ± 0.15
B3	500	10	67.41 ± 0.69	280.20 ± 1.98	0.499 ± 0.014	−28.01 ± 0.53
B4	100	30	41.15 ± 0.94	240.58 ± 7.61	0.576 ± 0.093	−26.70 ± 0.11
B5	300	30	62.11 ± 0.65	274.20 ± 1.80	0.544 ± 0.043	−27.35 ± 0.25
B6	500	30	76.12 ± 1.18	338.70 ± 4.33	0.728 ± 0.010	−28.25 ± 0.96
B7	100	50	45.03 ± 1.65	249.25 ± 0.75	0.773 ± 0.007	−27.70 ± 0.20
B8	300	50	64.32 ± 2.84	384.50 ± 0.50	0.538 ± 0.048	−28.20 ± 0.10
B9	500	50	81.89% ± 1.25	459.40 ± 4.09	0.645 ± 0.003	−28.70 ± 0.152

Note: Data represented as mean ± SD (n = 3). Abbreviations: BEO: bergamot essential oil, NPs: nano-phytosomes, PC: phosphatidylcholine, EE%: entrapment efficiency percentage, PS: particle size, PDI: polydispersity index, and ZP: zeta potential.

**Table 2 pharmaceuticals-16-00128-t002:** Output data of the two full factorial designs (3^2^) analysis of BEO-NPs formulations and predicted and observed values for the selected formula (B3).

Responses	EE%	PS (nm)	PDI	ZP (mV)
Adequate precision	42.34	64.44	6.44	9.97
Adjusted R^2^	0.99	0.99	0.56	0.82
Predicted R^2^	0.97	0.98	0.10	0.62
Significant factors	X_1_, X_2_	X_1_, X_2_	----	X_1_, X_2_
Predicted value of selected formula (B3)	67.02	281.5	0.541	−28.25
Observed value of selected formula (B3)	67.41	280.2	0.499	−28.01

**Table 3 pharmaceuticals-16-00128-t003:** The percentage reduction for the comedones, inflammatory lesions, and the total lesions for patients after application of the optimized BEO-NPs formulation on the right side of the face and BEO-NPs-SP formulation on the left side of the face.

		RT	LT	*p* Value
		N = 20	N = 20	
Com. B	Median	8	10.5	
	Mean ± SD	8.10 ± 3.09	9.65 ± 3.35	
	IQR	6.25–10.00	6.50–12.75	
	Range	2.00–14.00	4.00–14.00	
Com. A	Median	3	0	
	Mean ± SD	3.40 ± 2.26	1.20 ± 1.61	
	IQR	2.00–5.00	0.00–2.75	
	Range	0.00–8.00	0.00–5.00	
*p* value		*p* < 0.0001	*p* < 0.0001	
Com. Reduction %	Median	59.17	100	*p* < 0.0001
	Mean ± SD	63.10 ± 17.45	90.38 ± 12.53	
	IQR	50.00–74.11	78.15–100.00	
	Range	40.00–100.00	61.54–100.00	
Inf. B	Median	16.5	14	
	Mean ± SD	15.90 ± 2.10	13.75 ± 2.92	
	IQR	14.25–18.00	11.25–16.00	
	Range	11.00–18.00	9.00–18.00	
Inf. A	Median	6	1	
	Mean ± SD	5.85 ± 1.04	1.05 ±1.15	
	IQR	5.00–6.75	0.00–2.00	
	Range	4.00–8.00	0.00–4.00	
*p* value		*p* < 0.0001	*p* < 0.0001	
Inf. Reduction %	Median	64.29	93.31	*p* < 0.0001
	Mean ± SD	63.16 ± 4.93	93.37 ± 6.92	
	IQR	61.11–66.67%	87.85–100.00%	
	Range	50.00–70.59%	77.78–100.00%	
Tot. B	Median	24	23.5	
	Mean ± SD	24.00 ± 2.83	23.55 ± 4.77	
	IQR	22.00–25.75	20.50–27.75	
	Range	19.00–30.00	14.00–29.00	
Tot. A	Median	9	2	
	Mean ± SD	9.25 ± 1.86	2.25 ± 1.83	
	IQR	8.00–10.75	0.25–4.00	
	Range	6.00–13.00	0.00–6.00	
*p* value		*p* < 0.0001	*p* < 0.0001	
Tot. Reduction%	Median	61.39	92.59	*p* < 0.0001
	Mean ± SD	61.60 ± 5.24	91.28 ± 7.06	
	IQR	57.44–66.35	84.76–98.91	
	Range	52.38–70.83	77.78–100.00	

Abbreviations: BEO: bergamot essential oil, NPs: nano-phytosomes, SP: spironolactone, Inf.: inflammatory lesions, Com.: comedones, Tot.: total number of inflammatory lesions and comedones, A: after, B: before, RT: right side treatment, and LT: left side treatment.

**Table 4 pharmaceuticals-16-00128-t004:** Full factorial design for optimization of BEO-loaded NPs.

Factors (Independent Variables)	Levels	
X_1_: PC amount (mg)	100	300 500
X_2_: Cholesterol amount (mg)	10	30 50
**Responses (dependent variables)**	**Constraints**	
Y_1_: EE (%)	Maximize	
Y_2_: PS (nm)	Minimize	
Y_3_: PDI	Minimize	
Y_4_: ZP (mV)	Maximize	

Note: All formulations contain 300 mg BEO. Abbreviations: BEO: bergamot essential oil, NPs: nano-phytosomes, PC: phosphatidylcholine, EE%: entrapment efficiency percentage, PS: particle size, PDI: polydispersity index, and ZP: zeta potential.

**Table 5 pharmaceuticals-16-00128-t005:** Atomic composition of the constructed BEO-NPs-SP formulation simulated system.

Solvation State	Atomic Composition (№. of Atoms)										
	SP	PC	COL	LIM	LIN	BPI	GTP	GMC	BRG	Water	Total
100% Water	61	135	74	26	29	32	26	48	47	3 × 4393	13,657

Abbreviations: BEO: bergamot essential oil, NPs: nano-phytosomes, SP: spironolactone, PC: phosphatidylcholine, COL: cholesterol, LIM: limonene, LIN: linalool, BPI: β-pinene, GTP: γ-terpinene, GMC: 5-geranyloxy-7-methoxycoumarin, BRG: bergamottin.

## Data Availability

Data is contained within the article.

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
