# Peer review of "Exploring the Synergistic Effect of Bergamot Essential Oil with Spironolactone Loaded Nano-Phytosomes for Treatment of Acne Vulgaris: In Vitro Optimization, In Silico Studies, and Clinical Evaluation"

_pharmaceuticals, 2023, doi:10.3390/ph16010128_

Round 1

Reviewer 1 Report

The main target of the current work was to formulate and optimize a novel Bergamot  essential oil loaded nano-phytosomes and then combine it with Spironolactone in  order to clinically compare the efficiency of both formulations against acne vulgaris. The achieved results show the combination of two components loaded in Nano-phytosomes in mitigating the symptoms of acne vulgaris. However, minor revision should be needed.

1. Line 91,"in the current cstudy", please check the full paper for similar spelling error carefullly.

2..There were some faults in Table(2). In addition,the PDI of the measured BEO-NPs ranged from 0.41 to 0.77. Why were these PDI so big? 

Author Response

The reviewer's comments regarding manuscript editing have been modified and edited in the manuscript regarding the spelling mistakes.

For Table 2, 
I revised the content of table 2 and didn’t find the mistake.
However, kindly be informed that we have the selected formula which is B3 and its results is illustrated in table 2 
While after that we added Spironolactone to the selected formula and name it the optimized formula and its results is demonstrated in the manuscript and there was a difference between both results before and after addition of Spironolactone

Further, high PDI values are expected in nanocarriers prepared by thin film hydration, as this method produces a population with a polydisperse population.

Salama, A.H. and Aburahma, M.H., 2016. Ufasomes nano-vesicles-based lyophilized platforms for intranasal delivery of cinnarizine: preparation, optimization, ex-vivo histopathological safety assessment and mucosal confocal imaging. Pharmaceutical Development and Technology21(6), pp.706-715.

Reviewer 2 Report

The authors report a complex study on bergamot oil nanophytosomes combined with spironolactone for topical use in the treatment of acne vulgaris. The phytosomes were formulated by thin film hydration, characterized by entrapment, particle size, zeta potential, and morphology. The effect of the formulation was proven in clinical study.

The manuscript is well written and provides sufficient experimental details that support the conclusions. I recommend it to be accepted in present form.

Author Response

The authors would like to thank the reviewer for his/her encouragement and support.